# Identification of Various InDel-II Variants of the White Spot Syndrome Virus Isolated from Frozen Shrimp and Bivalves Obtained in the Korean Commercial Market

**DOI:** 10.3390/ani13213348

**Published:** 2023-10-27

**Authors:** Joon-Gyu Min, Hyun-Do Jeong, Kwang-Il Kim

**Affiliations:** Department of Aquatic Life Medicine, Pukyong National University, Busan 48513, Republic of Korea; cdmin0621@gmail.com (J.-G.M.);

**Keywords:** InDel-II, molecular epidemiology, transmission, WSSV

## Abstract

**Simple Summary:**

The white spot syndrome virus (WSSV) is a harmful pathogen with significant economic implications for the shrimp industry. This study investigated the presence of WSSV in frozen shrimp and bivalves in the Korean commercial market, with a specific focus on WSSV genetic variants. The research revealed that WSSV was detected in both domestic and imported shrimp, with higher viral loads in domestic samples. Various genetic variants of WSSV have been identified that exhibit differences in pathogenicity associated with longer deletion lengths in specific genetic regions linked to initial viral replication. Additionally, the presence of WSSV variants in bivalve mollusks suggested their potential utility as biomarkers for viral tracking. This study offers valuable insights into the prevalence and genetic diversity of WSSV in aquatic animal products that are essential for disease control and management in the aquaculture industry.

**Abstract:**

White spot syndrome virus (WSSV) poses a significant threat to the global shrimp industry. We investigated the presence of WSSV in frozen shrimp (*n* = 86) and shellfish (*n* = 185) from the Korean market (2010–2018). The detection rate of first-step polymerase chain reaction (PCR) in domestic shrimp was 36.8% (7/19), whereas that in imported shrimp was 0.01% (1/67). Furthermore, the WSSV genome was amplified from domestic bivalve mollusks by first- and second-step PCR with accuracies of 3.4% (5/147) and 15.6% (23/147), respectively. The genetic relatedness of InDel-II regions among WSSVs detected in domestic shrimp groups revealed four variants (777, 5649, 11,070 and 13,046 bp insertion or deletion), and imported shrimp groups had four variants (10,778, 11,086, 11,500 and 13,210 bp) compared with the putative ancestor WSSV strain. The 5649 bp variant was the dominant type among the WSSV variants detected in domestic shrimp (54.5%, 6/11). Notably, bivalve mollusks exhibited six variants (777, 5649, 5783, 5876, 11,070 and 13,046 bp), including four variants detected in shrimp, indicating that bivalve mollusks could facilitate WSSV tracking. In a challenge test, whiteleg shrimp (*Litopenaeus vannamei*) exhibited varying mortality rates, indicating a link between InDel-II deletion and viral replication. These findings highlight the complexity of WSSV transmission.

## 1. Introduction

Globally, there is an increasing trend in the cross-border trade of aquatic animals and their products, implying potential disease transmission through movement between different countries. White spot syndrome virus (WSSV), which belongs to the family *Nimaviridae*, is one of the most problematic viruses infecting shrimp, including whiteleg shrimp (*Litopenaeus vannamei*), leading to serious economic losses in the shrimp culture industry worldwide. An outbreak of WSSV was detected in a previously disease-free region of the Logan River near Queensland, Australia, where WSSV was primarily detected in harvested shrimp [1]. This marked a significant shift, as Australia had previously been recognized as a disease-free region. The reported losses from this incident amounted to approximately 87 million Australian dollars (www.daf.qld.gov.au, accessed on 29 August 2023). The anticipated pathway of WSSV introduction into Australia has been associated with the import of infected bait shrimp [2]. Additionally, WSSV has been reported in imported frozen shrimp in Korea [3,4].

The potential routes of introduction for specific aquatic animal diseases involve susceptible hosts and vectors or carriers within the aquatic environment [5,6,7]. Infectious viruses, such as WSSV, can be transmitted to host organisms through vectors or carrier organisms [8,9]. Bivalve mollusks, such as oysters, mussels and clams, have filter-feeding activity, a process through which they ingest various particles, including viral particles, present in the surrounding water, leading to their accumulation within their tissues [10]. Previous studies have reported that viral particles accumulate in oysters and clams. Particularly, viral particles within oyster tissues, including the digestive gland, can remain specifically attached and are not completely inactivated by digestive enzymes [11,12]. Moreover, there have been reports of the detection of the barfin flounder nervous necrosis virus genotype, previously found exclusively in foreign countries and not in domestic fish, in bivalve mollusks [13]. These findings contribute to our understanding of aquatic virus transmission pathways in marine environments. Particularly, one study [14] demonstrated the accumulation of WSSV in oysters placed in the water intake canals of whiteleg shrimp farm facilities. Furthermore, they emphasized the potential use of WSSV detection as a bioindicator for bivalve mollussks, even before the onset of the disease in shrimp farms [15].

In molecular epidemiology, elucidation of the genetic characteristics of the causative agent of a disease can serve as a useful tool for understanding the pathways of its introduction and transmission. Marks et al. [16] performed a genetic sequence analysis of WSSV isolates from Taiwan (WSSV-TW), China (WSSV-CN) and Thailand (WSSV-TH). They identified several major variable genomic locations, including (i) a large genomic insertion or deletion (InDel) region within the encoding ORF14/15 (InDel-I) and ORF23/24 (InDel-II), (ii) a variable number of tandem repeats (VNTRs) in ORF75, ORF94 and ORF125, (iii) a genomic region encoding transposase and (iv) single-nucleotide polymorphisms (SNP). Among these, ORF 23/24, which represents an InDel-II region with an identified deletion length of approximately 13 kb, is a significant genetic marker for epidemiological tracking [17]. Consequently, numerous studies have utilized the InDel region to investigate the transmission pathways of WSSV between regions or on global scales, aiming to ascertain the sources of WSSV introduction [18,19]. Marks et al. [20] suggested that differences in the sizes of the InDel region may lead to variations in pathogenicity. Based on the hypothesis that evolutionarily redundant genes may be gradually deleted from the genome over generations, leading to a reduction in the overall genome size, the authors demonstrated the significance of the InDel-II region as a molecular epidemiological marker.

In the present study, we investigated the presence of WSSV in domestic and imported frozen shrimp and bivalves. The detected WSSV samples were analyzed for a molecular epidemiological marker (InDel-II region), and genetic relatedness and pathogenicity were determined.

## 2. Materials and Methods

### 2.1. Samples

Frozen shrimp and bivalves were collected to investigate the presence of WSSV and its genetic correlation, with a particular focus on examining the genetic variations in the viral isolates found in these samples. Five subcuticular connective tissues from shrimp and three digestive glands from shellfish were analyzed within each group of randomly selected individuals. At the detection stage, if any shrimp or shellfish tested positive, the entire group was considered a positive sample. For frozen shrimp, 19 groups of whiteleg shrimp produced on domestic farms were collected from 12 regions, and 67 groups of whiteleg shrimp imported from various countries, including Thailand, Malaysia, Ecuador, Indonesia, Vietnam, Argentina, Saudi Arabia and New Zealand, were purchased from a commercial market in Korea between 2011 and 2018 (Appendix A). For the bivalve samples from 10 domestic regions, 147 groups of bivalves displaying apparent health were collected from commercial markets around the coast between 2010 and 2014 (37 Pacific oysters [*Crassostrea gigas*], 50 mussels [*Mytilus edulis*], 23 Manila clams [*Venerupis philippinarum*], 14 granular arks [*Tegillarca granosa*], 12 Venus clams [*Mercenaria mercenaria*], 5 common orient clams [*Meretrix meretrix*] and 6 scallops [*Patinopecten yessoensis*]). Furthermore, for imported bivalves, samples were collected from four different regions in China (13 Manila clams, 6 Venus clams, 4 common orient clams, 2 scallops, 3 cyclina clams [*Cyclina sinensis*] and 3 other shellfish) and two different regions in Japan (8 granular arks), and they were sampled at an airport or harbor before exposure to Korean seawater (Appendix A).

### 2.2. Polymerase Chain Reaction (PCR) Primer Design

For WSSV detection and determination of the InDel-II region, PCR primers were designed for the VP 28 protein and ORF 23/24, respectively (Table 1). For WSSV detection, a nested PCR protocol was designed targeting the *VP28* gene. In the first and second steps, VP28-F1/R1 and WSSV VP28-F2/R2 primers were used, respectively. Additionally, to determine the InDel-II ORF 23/24 region, eight nucleotide sequences of reported WSSV strains were obtained from GenBank (Bethesda, MD, USA), including those from Taiwan (WSSV-TW; GenBank Access. No. AF440570), China (WSSV-CN and WSSV CN01; Access. No. AF332093 and KT995472, respectively), Korea (K-LV1; Access. No. JX515788), Mexico (MEX2008; Access. No. KU216744), India (WSSV-IN; Access. No. EU327500), and Ecuador (WSSV-EC-15098; Access. No. MH090824) and Thailand (WSSV-TH; Access. No. AF369029). Sequence alignment revealed that the InDel-II region could be categorized into three types based on the deletion size, and three primer sets (ORF 23/24 -S, -M and -L) were designated based on this alignment. Briefly, PCR targeting of the InDel-II region of WSSV was performed in a three-step process. Amplification was performed using the InDel-II-S primers. If an amplicon was not detected in the initial PCR step, a second amplification step was performed using the InDel-II-M primer. If an amplicon remained undetected, a third amplification step was performed using the InDel-II-L primer (Appendix A).

### 2.3. Viral DNA Extraction and WSSV Detection

Ten milligrams of each tissue was homogenized using a mini-homogenizer (Pellet Pestle Motor Cordless, Sigma-Aldrich, St Louis, MO, USA) with phosphate-buffered saline (PBS, Sigma-Aldrich). Total viral DNA was extracted using the Exgene Tissue SV Mini Kit (Gene All Biotechnology, Seoul, Republic of Korea) according to the manufacturer’s instructions. In the first and second steps of nested PCR amplification, the final volume of the reaction mixture was 20 µL, containing 10 µL of ExPrime Taq Premix (2x) (Genet Bio, Daejeon, Republic of Korea), 1 µM of each primer, 1 µL of DNA or first-step PCR amplicon, and distilled water. The amplification conditions were as follows: 94 °C for 5 min, followed by 40 cycles of 94 °C for 30 s, 60 °C for 15 s and 72 °C for 20 s and a final extension at 72 °C for 7 min in an Applied Biosystems^®^ (Foster City, CA, USA) 2720 thermal cycler. The amplified products were analyzed via 1% agarose gel electrophoresis. To avoid cross-contamination in PCR, WSSV-free shrimp tissues were used as negative controls during sample analysis.

### 2.4. Sequence Analysis of InDel-II Region

To determine the InDel-II region of WSSV detected in shrimp and bivalves, PCR targeting the WSSV ORF 23/24 region was performed on positive samples from the first- and second-step PCRs, following the protocol described above. Then, PCR amplicons were purified using agarose gel electrophoresis and Expin™ PCR SV mini (GeneAll Biotechnology, Seoul, Republic of Korea) and cloned into the pGEM-T Easy Vector System I (Promega, Fitsburg, WI, USA) according to the manufacturer’s instructions. Each purified plasmid was sequenced using an ABI3730 Genetic Analyzer (Applied Biosystems, Waltham, MA, USA) and analyzed using Finch TV version 1.2.0 (Geospiza, Seattle, WA, USA). Sequences were compared via gene alignment using the BioEdit software (ver. 7.0.6, North Carolina State University, Raleigh, NC, USA) and the MACAW program to compare the size of the InDel-II region with that of the putative ancestral WSSV strain (WSSV-TW) [16]. 

### 2.5. Quantitative PCR

The number of viral genome copies in the tissue was determined using a Rotor Gene 6000 thermal cycler (Qiagen, Germany) according to the manufacturer’s guidelines. Quantitative PCR (qPCR) was performed in a 20 µL reaction mixture containing 10 µL of TOPreal™ qPCR Premix (2×) (Sybr Green with low ROX, Enzynomics, Daejeon, Korea), qPCR primer set at a concentration of 500 nmol/L each, 1 µL of DNA, and distilled water. The amplification conditions were as follows: 95 °C for 15 min, followed by 40 cycles of 95 °C for 10 s, 60 °C for 15 s and 72 °C for 20 s. Melting curves were recorded by plotting fluorescence signal intensity versus temperature. As a positive control, recombinant plasmids containing 281 bp from the VP 28 region of WSSV were replicated from the transformed Escherichia coli DH5α strain. A serial 10-fold dilution of the control plasmid was used to establish a standard curve (1.0 × 10^7^ copies/µL to 1.0 × 10^1^ copies/µL). All samples used in this study were tested in duplicate, and the viral genome copies for each group were calculated as the mean copy values from each individual. Significant differences in WSSV genome copies between domestic and imported shrimp groups were determined with Student’s *t*-test using GraphPad Prism (Ver. 9.5.0; GraphPad, Boston, MA, USA) software. The level of significance was set at *p* < 0.05.

### 2.6. Pathogenicity of WSSV InDel-II Region Variants

Based on the analysis of the WSSV InDel-II region in domestic shrimp, four variants (777, 5649, 11,070 and 13,046 bp) were identified. To investigate the pathogenicity of WSSVs with different genome deletion sizes, a challenge test was conducted in whiteleg shrimp. Among the four variants, the representative samples that tested positive in the first PCR step were selected. Representative samples were designated as Kr-1, Kr-2, Kr-3 or Kr-4 isolates and were used for the challenge test (Figure 1). Additionally, to assess the pathogenicity of the WSSVs detected using first-step PCR in bivalves, homogenates of midgut tissue were used as the inoculum. Whiteleg shrimp (mean body weight, 15.4 ± 2.3 g) were obtained from a whiteleg shrimp farm located in Geoje, Gyeongsangnam-do, Republic of Korea, and they were confirmed to be WSSV-free via PCR described above. In a preliminary experiment for the activation of WSSVs, each tissue homogenate (10^6^ genome copies/shrimp) was injected into five healthy whiteleg shrimp and kept in 50 L tanks for 10 days at 23 °C. Abdominal muscle tissues from dead or moribund shrimp were used as inocula to determine the pathogenicity of the InDel-II variants. The WSSV inoculum was prepared using a modified method described previously [22]. The abdominal muscle tissue was homogenized in PBS (1:9, *w*/*v*). After centrifugation at 8000× *g* for 10 min, the supernatant was filtered through a 0.45 µm syringe filter and aseptically transferred to sterile micro-centrifuge tubes. The viral genome copies of each inoculum were determined via qPCR as described above. To compare the WSSV pathogenicity between InDel-II variants, the whiteleg shrimp were intramuscularly injected with 100 µL of WSSV suspension at 10^5^ WSSV genome copies/shrimp (*n* = 12). PBS was injected into the shrimp as a negative control. After the challenge, the shrimp were maintained in a 50 L tank at 23 °C for 10 days while being examined for clinical signs of WSSV infection and mortality. DNA from the abdominal muscles of dead shrimp was extracted and tested for WSD (white spot disease) via PCR. Significant differences in WSSV mortality between variants were determined via a log-rank test using the aforementioned software.

Furthermore, additional pathogenicity experiments in the early-infection stage were conducted using the Kr-1 and Kr-4 variants, which exhibited significant differences from previous pathogenicity tests. To perform these experiments, we employed groups, each consisting of 20 individuals, and injected them with the same WSSV copy value (1.0 × 10^6^ copies/shrimp) using two variants (Kr-1 and Kr-4 variants) following the aforementioned conditions and methods. Subsequently, three live shrimp at each time point (12, 24, 48 and 72 h post injection) were used for the analysis. For the statistical analysis, WSSV copy values were compared using a two-way ANOVA. The level of significance was set at *p* < 0.05.

## 3. Results

### 3.1. Detection of WSSV in Frozen Shrimp and Bivalve Mollusks

Domestic and imported shrimp samples accounted for 57.9% (11/19) and 47.8% (32/67) of second-step PCR samples, respectively (Table 2). In first-step PCR, 36.8% (7/19) of the domestic samples tested positive, whereas in the case of imported samples, only 1 group out of 67 tested positive, resulting in a detection rate of 1.49%. In the case of domestic bivalve samples, 23 of 147 bivalves were identified as WSSV-positive in second-step PCR (15.6%), among which five species were confirmed to be positive in first-step PCR (3.4%) (Table 3). For imported bivalves, WSSV was detected in only one sample of Venus clams from China during first-step PCR (3.3%, 1/30). In second-step PCR, WSSV was detected in Venus clams, common orient clams, Chinese cyclina from China (10.0%, 3/30) and granular arks from Japan (12.5%, 1/8) (Table 4).

### 3.2. Quantification of WSSV

Standard curves were generated using serial 10-fold dilution of the control plasmid. The mean data from experiments conducted in triplicate were used, and an R^2^ value of 0.99967 was obtained, indicating a strong linear relationship between the Ct values. Of the first and second PCR-positive shrimp samples (11 domestic and 32 imported shrimp), viral genome copies were determined in domestic (*n* = 11) and imported shrimp (*n* = 15) (Table 3, Table 4 and Appendix A). The quantification of WSSV in a total of 11 WSSV-positive shrimp from domestic farms showed 2.24 × 10^2^–8.89 × 10^6^ WSSV genome copies/mg (mean, 3.06 × 10^6^ WSSV genome copies/mg) (Figure 2, Appendix A). Even though 32 imported shrimp showed PCR-positive results, viral genome copies were determined from 15 samples with a mean of 7.34 × 10^2^ WSSV genome copies/mg (range, 1.46 × 10^1^–1.07 × 10^4^ WSSV genome copies/mg), which significantly differed from that of domestic shrimp (Figure 2, Appendix A). The quantitative analysis was also conducted on PCR-positive samples from 23 domestic and 4 imported bivalves. Among the domestic bivalve samples, nine samples were amplified and showed a range from 5.92 × 10^2^ to 8.86 × 10^4^ WSSV genome copies/mg (mean, 2.39 × 10^4^ WSSV genome copies/mg) (Appendix A). In the case of imported bivalves, only one sample (sample code 13-CHN-Vc-3) showed amplification with a genome copy value of 4.17 × 10^4^ WSSV genome copies/mg in the Chinese Venus clam.

### 3.3. WSSV InDel-II Variants from Frozen Shrimp and Bivalve Mollusks

In discriminatory PCR, WSSV variants were amplified from all 11 shrimp groups in the domestic samples (first- and second-step PCR positives), whereas in the case of imported samples, only four variants were amplified among the 32 samples. There were no differences in the variants among the individuals within each group. Sequence analyses confirmed that products amplified by each primer set represented four deletion types, including 777 bp (*n* = 1), 5649 bp (*n* = 6), 11,070 bp (*n* = 3) and 13,046 bp (*n* = 1) in length, compared to the same parameters associated with WSSV-TW (Figure 1; there were no differences among individuals within each group). In contrast, among the 32 imported shrimp samples, amplification was observed in only four variants, each revealing a different deletion type: 10,778 bp (sample code: 17-VNM-1), 11,086 bp (sample code: 15-ECU-3), 11,500 bp (sample code: 17-ECU-3) and 13,210 bp (sample code: 15-THA-2) (Appendix A). However, the results for the other samples were inconclusive because of the absence of a distinct band or the presence of multiple bands, making it difficult to accurately identify the deletion types. In the case of bivalve samples, amplification was observed in only 8 of the 23 domestic samples, revealing the presence of six deletion types. In addition to the four types identified in domestic shrimp samples, variants with deletion sizes of 5873 or 5876 bp were identified in domestic bivalve samples (Figure 1).

### 3.4. Pathogenicity of Different InDel-II Variants

Using the four activated WSSV variants described above, the challenge tests were conducted by injecting the same number of genome copies (10^6^ WSSV genome copies/shrimp). Although the overall cumulative mortality was consistently 100%, variations in the rate of mortality progression were observed depending on the deletion type. Kr-4 variants, characterized by longer deletion sizes, exhibited faster mortality progression than that exhibited by Kr-1 variants, which have shorter deletion sizes. The LT_50_ (median lethal time) for the Kr-4 variants was 3 days, whereas that for the Kr-1 variants was 5 days (Figure 3). Furthermore, the log-rank test revealed a significant difference in cumulative mortality progression between the Kr-1 and Kr-4 variants. However, no mortality was observed in the group inoculated with the WSSVs detected in the bivalves.

In an additional experiment comparing the pathogenicity of Kr-1 and Kr-4 variants during the initial stages of infection, the quantification of Kr-1-variant-injected shrimp at 12 hpi showed 2.87 × 10^2^–7.48 × 10^2^ WSSV genome copies/mg (mean, 5.17 × 10^2^ WSSV genome copies/mg). In contrast, Kr-4-variant-injected shrimp exhibited a significantly higher viral load, ranging from 1.03 × 10^5^ to 2.66 × 10^6^ WSSV genome copies/mg (mean, 1.38 × 10^6^ WSSV genome copies/mg), found to be statistically significant (*p*-value = 0.0385) between the variants at 12 hpi (Figure 4). Furthermore, at 24 hpi, the Kr-1 variants showed an average viral copy value of 1.66 × 10^4^ WSSV genome copies/mg, and the Kr-4 variants exhibited a higher viral copy value at 1.84 × 10^7^ WSSV genome copies/mg. The difference between the variants was found to be statistically significant (*p*-value = 0.0359).

## 4. Discussion

Disease transmission in aquatic environments involves infected animals as well as pathogen-contaminated feed, bait and water [23]. These factors contribute to the introduction of infectious agents into aquaculture facilities, increasing the likelihood of disease establishment and spread among susceptible organisms. Notably, despite the understanding of WSSV transmission in previous studies, global disease outbreaks caused by WSSV in a diverse range of hosts and carriers/vectors remain a significant concern. WSSV exhibits genetic variations in specific gene regions, such as the VNTRs and InDel regions. Among these, the InDel region serves as a valuable molecular marker for epidemiological studies on regional and global scales [18]. The objective of this study was to investigate the presence of WSSV in frozen shrimp and bivalves, and to determine genetic variants based on the sequence deletion pattern of the InDel-II region. Our findings highlight the need for molecular epidemiological analyses to assess the potential transmission of WSSV through frozen shrimp and bivalve products in the Korean commercial market. In particular, WOAH [24] suggested that a positive result in first-step PCR indicates a serious infection stage, whereas a positive result in second-step PCR indicates a latent or carrier-state infection stage. Accordingly, in this study, second-step PCR was conducted to confirm the carrier states of frozen shrimp and bivalves. Furthermore, to distinguish potential cross-contamination that could easily occur during second-step PCR, as highlighted by Claydon et al. [25], we employed WSSV-free whiteleg shrimp tissues as controls. This approach ensured that the results obtained, which utilized genetic comparisons of the InDel-II region of WSSV found in these products, could be attributed to the presence of WSSV in frozen shrimp and bivalves, enhancing the accuracy of our analysis and allowing us to evaluate the likelihood of transmission.

The WSSV detection rates determined using second-step PCR in domestic and imported frozen shrimp were similar, at 57.9% (11/19) and 47.8% (32/67), respectively. However, the detection rate of first-step PCR in domestic shrimp was 36.8% (7/19), which was significantly higher than that in the imported samples (3.1%, 1/32) (Table 2). Moreover, the average WSSV copy number detected in domestic shrimp was 3.06 × 10^6^ viral genome copies/mg, which was approximately 3000 times higher than that detected in imported shrimp (8.99 × 10^2^ viral genome copies/mg) (Figure 2). Furthermore, although WSSVs from domestic shrimp retained their pathogenicity, those from imported shrimp did not induce infection in whiteleg shrimp. Although WSSV has been shown to maintain its pathogenicity when stored in a frozen state within the range of −20 to −70 °C for 2 years [26,27], it is essential to acknowledge the possibility of prolonged storage, owing to the nature of frozen products, and domestic shrimp have a shorter farm-to-consumer distribution period compared to that associated with imported products. Additionally, Kim et al. [7] explored the correlation between the infection stages proposed by Lightner [28]. The results of this study showed that WSSV-infected shrimp in the G1 stage (range, 7.2 × 10^2^ to 9.9 × 10^3^ viral genome copies/mg) did not induce mortality. Overall, the differences in the WSSV detection rates, number of viral copies and mortality induction between domestic and imported shrimp may be attributed to differences in the infection grade (or level), distribution process (duration) and storage conditions.

For the bivalve samples, including 147 from domestic sources, 30 from China and 8 from Japan, the WSSV detection rates from second-step PCR were 15.6, 10.0 and 12.5%, respectively. Although second-step PCR yielded positive results from bivalve samples (*n* = 27), viral genome copies could be determined in only 10 samples (9 from domestic bivalves and 1 from an imported bivalve). In a previous study, the 95% limit of detection between second-step PCR and qPCR was 6.8 times different (0.70 viral genome copies/rxn for second-step PCR compared to 4.67 viral genome copies/rxn for qPCR) [29]. Furthermore, the high carbohydrate content in the digestive gland tissues of bivalves can interfere with PCR amplification [30]. Thus, WSSV in the bivalve samples may be near the detection limit of qPCR, or inhibitor substances, such as high carbohydrates found in the digestive gland used for DNA extraction in this study, could potentially interfere with the determination of viral genome copies via qPCR.

According to the detection rates in various species, the detection rate of WSSV in tissues did not differ, and WSSV may be deposited mainly via physical rather than chemical mechanisms. A previous study [22] reported the absence of a quantitative RNA increase in *Meretrix lusoria* following WSSV immersion, supporting the proposition that WSSV within shellfish does not actively replicate, suggesting that shellfish might primarily act as a carrier. Similarly, previous studies on Pacific oysters (*Crassostrea gigas*) placed in a shrimp farm water supply canal have suggested a potential WSSV bioindicator [15]. Although feeding shrimp with common oriental clams immediately accumulated WSSV and could induce infection [22], WSSVs detected via first-step PCR in bivalves did not induce mortality in whiteleg shrimp. This suggests the possibility of viral inactivation or an insufficient amount of the virus causing the infection. Further research is required to determine the relationship between WSSV accumulation in bivalves and viral survival rates.

Marks et al. [16] analyzed three fully sequenced WSSV strains (WSSV-TW; Access. no. AY753327; WSSV-CN, Access. no. AF332093; and WSSV-TH, Access. no. AF369029), indicating that WSSV has diverse polymorphic genomic loci. Among these, “variable region ORF 23/24” exhibited the largest deletion, with deletions reaching approximately 13 kb. In the fields of evolutionary biology and protein engineering, the importance of InDel analysis has been growing alongside mutations [31]. InDels play crucial roles in introducing genetic diversity and unlocking new protein functionalities. Scientists are employing combinatorial strategies, guided by phylogenetic and structural analyses, to introduce InDels into genetic sequences. This approach aims to enhance the catalytic specificity of enzymes and to improve the binding affinity of engineered antibodies. This underscores the potential for exciting innovations in the field of protein science [32]. Zwart et al. [17] observed significant statistical correlations between the year of occurrence and the size of the deletion in the variable ORF 23/24 region, also known as the InDel-II region. They proposed that this insight could be leveraged to investigate genetic relatedness and shed light on WSSV transmission pathways. Variants of the WSSV InDel-II region have been reported in countries, such as India, Vietnam, Madagascar and Brazil [33,34,35]. Based on the sequence deletion pattern in the InDel-II region, four variants of WSSVs from domestic and imported shrimp were identified (777, 5649, 11,070 and 13,046 bp insertions or deletions from domestic shrimp with deletions of 10,778, 11,086, 11,500 and 13,210 bp of that region, respectively). Among the WSSV variants, the 5649 bp deletion type was dominant in Korea (54.5%, 6/11) (Figure 1) and was closely related to the WSSV strain (WSSV-KR) reported in Korea in 2011 [36]. The exact mechanism underlying the change in the InDel region of WSSV over generations has not been conclusively elucidated. However, considering the similarity in the deletion size observed in variants from a previous report [37] and those identified in this study, it is plausible to conclude that WSSV-causing outbreaks in Korea are likely to persist and inflict damage to the domestic shrimp industry. Nevertheless, WSSVs from frozen shrimp imported from Ecuador (sample code: 15-ECU-3) and Thailand (sample code:15-THA-2) in 2015 (Figure 1, Appendix A) exhibited deletion sizes that matched those of the reference strains from Ecuador and Thailand (MH090824 and AF369029, respectively). Although these variants did not cause infection, the match of the InDel-II variant isolated from imported shrimp to the reference sequence of the exporting country suggests the possibility of introducing diverse WSSV types through trade, some of which may exhibit different pathogenicity. Notably, apart from the four variants detected in domestic shrimp, WSSVs from bivalves were also found to harbor three additional variants (5783, 5876 and 13,046 bp). The discovery of WSSV variants in bivalves, not identified in shrimp, indicates the potential risk for novel disease transmission dynamics. Furthermore, previously unreported SNPs were observed in the two WSSV variants found in bivalves (12-BS-Mu-9 and 14-GJ-Mu-2) at position 12,725 of the WSSV-TW sequence (GenBank access. no. AY753327). These unique variants could potentially be transmitted to shrimp or other susceptible species, thereby causing new patterns of disease outbreaks and uncharted genetic diversity within WSSV. This highlights the importance of continuous surveillance and genetic characterization for comprehensive understanding and strategizing for disease control and prevention in aquaculture. However, the samples of first- and second-step PCR positives were not amplified in the InDel-II region. This phenomenon may attributed to the very low copy number of WSSV, which corresponds to patterns identified in previous studies [38].

Overall, these findings indicate the potential for the introduction of exotic types of WSSV into Korea through imported frozen shrimp and bivalves. Although it remains unclear whether WSSV variants from bivalve mollusks represent novel WSSV strains or whether they result from deletions occurring during the transmission process, our results suggest the potential use of bivalve mollusks for the epidemiological tracking of WSSV.

Despite the lack of information on the factors affecting genetic deletions in viruses, it has been hypothesized that viruses evolve as generations progress, potentially favoring directions that confer advantages for replication speed [17]. Therefore, we challenged WSSV-free shrimp with four variants of WSSV based on the InDel-II region to investigate their respective pathogenicities. Although the cumulative mortality rate was 100.0% across all groups, Kr-4 variants, which have longer deletion lengths in the InDel-II region, exhibited faster progression of mortality than that exhibited by Kr-1 isolates with shorter deletion lengths (Figure 3). Additionally, additional experiments to determine pathogenicity in the early stages of WSSV infection were conducted with the Kr-1 and Kr-4 variants. The quantification of viral copy values within shrimp tissues revealed differences between the two variants at both 12 and 24 hpi, which were found to be statistically significant (*p*-value = 0.0385 at 12 hpi and *p*-value = 0.0359 at 24 hpi). These findings are consistent with those of a study by Krell [38], which suggested that a virus with a smaller genome size provides an advantage for replication. Additionally, in the InDel-II regions of the ancestral strains, two ORFs (*WSSV006* and *WSSV025*) are adjacent to *WSSV004*, an essential early viral gene [39]. This suggests that variations in the InDel-II region lead to differences in the early stages of viral replication, ultimately resulting in variations in pathogenicity. Based on these findings, it is reasonable to conclude that the observed differences in pathogenicity between the Kr-1 and Kr-4 variants could be attributed to variations in the InDel-II region.

## 5. Conclusions

WSSV was detected in both shrimp and bivalves in the Korean commercial market, with notably higher levels in domestic shrimp, as observed in the quantitative analysis. The sequence deletion pattern in the InDel-II region revealed WSSV variants (four from domestic shrimp, four from imported shrimp and six from bivalve mollusks), indicating the possibility of the circulation of different WSSV variants in Korea. Differences in pathogenicity were observed among the four variants found in domestic shrimp, which may be attributed to differences in their initial replication capabilities. Notably, the WSSV variants found in bivalves also include those identified in shrimp, suggesting that bivalves could potentially serve as biomarkers for tracking WSSV. Overall, shrimp, known as hosts, and mollusks, recognized as carriers, allow the identification of various WSSV variants. This has significant implications for the epidemiological tracking of the virus source and for the distribution of WSSV types in endemic regions (or countries). Furthermore, it can be instrumental in estimating pathogenicity based on information from these areas. Further research is required to assess the viability of bivalve-accumulated WSSV.

## Figures and Tables

**Figure 1 animals-13-03348-f001:**
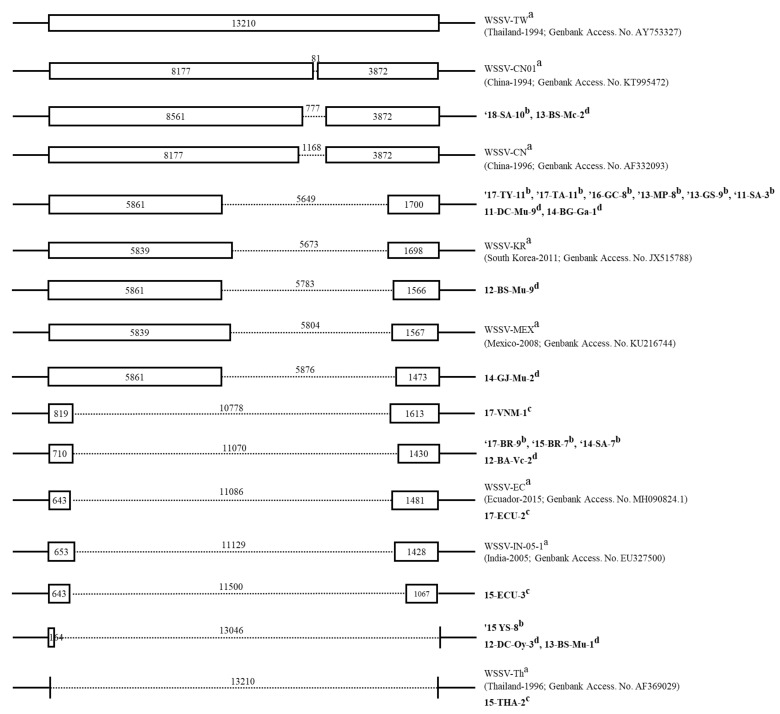
Schematic diagram of the InDel-II variable region of various WSSV variants. Sample codes of WSSV variants isolated in this study are in bold. The numbers in boxes or sequences indicate the length of the fragments. ^a^: WSSV variants from reference isolates in GenBank; ^b^: WSSV variants from domestic frozen shrimp; ^c^: WSSV variants from imported frozen shrimp; ^d^: WSSV variants from domestic bivalve samples.

**Figure 2 animals-13-03348-f002:**
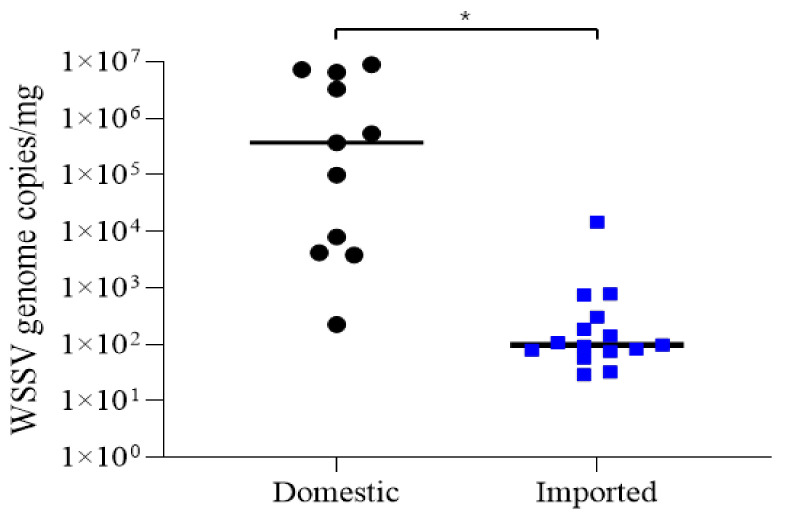
Comparison of WSSV genome copies in domestic and imported frozen shrimp. Each symbol represents mean values from five extractions. Black circles represent viral genome copies in tissues in domestic frozen shrimp. Blue circles represent viral genome copies in tissues in imported frozen shrimp. (** p* < 0.05; *t*-test).

**Figure 3 animals-13-03348-f003:**
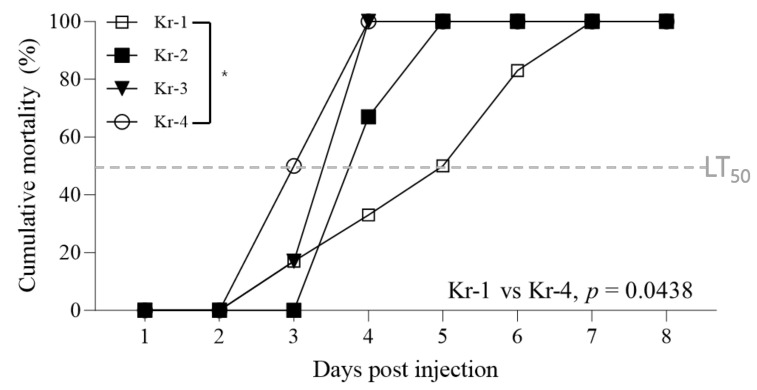
Cumulative mortality of whiteleg shrimp with different virus strains and challenged with each WSSV strain of 1.00 × 10^5^ WSSV genome copies/shrimp (*n* = 12). Each isolate (Kr-1, Kr-2, Kr-3 and Kr-4) was a representative sample that tested positive in first-step PCR. The horizontal gray broken line indicates LT_50_ (median lethal time) values. Statistical data were obtained using a log-rank test (* *p* < 0.05).

**Figure 4 animals-13-03348-f004:**
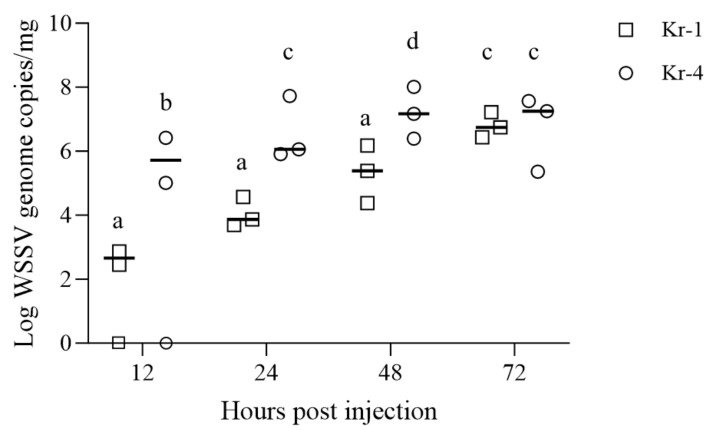
Viral copies of pleopods in *L. vannamei* (*n* = 20 each) injected with Kr-1 and Kr-4 variants (1.00 × 10^6^ WSSV genome copies/shrimp) at different interval times (12, 24, 48 and 72 hpi). Letters indicate differences among WSSV variants and hpi based on a two-way ANOVA (*p* < 0.05).

**Table 1 animals-13-03348-t001:** Primers used in this study for detection and the variable loci of WSSV.

Target	Primers	Sequence (5′–3′)	Positions *	Object (Expected Size, bp)	Reference
VP 28	W VP28 F1 * W VP28 R1	CTT TCA CTC TTT CGG TCG TGT C TCG GTC TCA GTG CCA GAG TA	278,875–278,896 279,456–279,475	First-step PCR (601)	This study
	WSSV VP28 F2 WSSV VP28 R2	CAC TGT GAC CAA GAC CAT CG GGT GCC AAC TTC ATC CTC ATC	278,957–278,976 279,354–279,374	Second-step PCR (408)	[8]
	WSSV qF WSSV qR	TGT GAC CAA GAC CAT CGA A CCA CAC CTT GAA TGT TC	278,960–278,978 279,223–279,239	Quantification (281)	[21]
ORF 23/24 (InDel-II region)	S—F1 S—R1	TAC ATG GGA GGG AGA GGT GAT TGC GAA ATA CGG GCA ATG TTT	10,457–10,477 11,982–12,005	First Discrimination First-step PCR	This study
	S—F2 S—R2	TCT GGG GCG CTT GTT ACT TG AAG GAG GAG GTG TTG GAG CTA	10,515–10,534 11,936–11,956	First Discrimination Second-step PCR	
	M—F1 M—R1	CGC CAG TAC CTT CTT CCA CT TCT CAA GGA GGA GAG AGC GT	8112–8131 14,726–14,745	Second Discrimination First-step PCR	
	M—F2 M—R2	GTC GAC AGG GAC TTC AAT ACC GTG TTG GTA AAT GCA CG	8185–8202 14,619–14,638	Second Discrimination Second-step PCR	
	L—F1 L—R1	CCA CTA GCC TTC CAC GTG TT GCA GTC GGC AAC ATC TTG TG	2208–2227 16,367–16,386	Third Discrimination First-step PCR	
	L—F2 L—R2	GTG CCC TTT TGC AAG GCA TA ATA CCG GCG AGT CTT GAA CC	2467–2486 16,180–16,199	Third Discrimination Second-step PCR	

* Primer sequence positions based on the WSSV-TH reference sequence (GenBank access. no. AF440570); PCR, polymerase chain reaction; WSSV, white spot syndrome virus.

**Table 2 animals-13-03348-t002:** Prevalence of white spot syndrome virus in tissues of domestic and imported frozen shrimp by nation.

PCR	Domestic (Korea)	Imported							
		Thailand	Malaysia	Ecuador	Indonesia	Vietnam	Argentina	Saudi Arabia	New Zealand
First-step PCR	7/19 * (36.8) **	1/19 (5.3)	0/12 (0.0)	0/12 (0.0)	0/10 (0.0)	0/6 (0.0)	0/4 (0.0)	0/2 (0.0)	0/2 (0.0)
Second-step PCR	11/19 (57.9)	8/19 (42.1)	7/12 (58.3)	7/12 (58.3)	4/10 (40.0)	4/6 (66.6)	0/4 (0.0)	2/2 (100)	0/2 (0.0)
Total	11/19 (57.9)	32/67 (47.8)							

* No. of positive groups/No. of all analyzed groups. ** Percentage of positive groups.

**Table 3 animals-13-03348-t003:** Prevalence of white spot syndrome virus (WSSV) in shellfish obtained from the market in South Korea.

Samples	Location	No. of Groups	No. of PCR Positives (%)	
			First-Step PCR	Second-Step PCR
Pacific oyster	BS *	25	0 (0.0) **	4 (16.0)
*(Crassostrea gigas)*	DC	12	0 (0.0)	2 (16.7)
Mussel	DC	21	2 (9.5)	3 (14.3)
(*Mytilus edulis*)	BS	14	1 (7.1)	2 (14.3)
	GJ	8	0 (0.0)	1 (12.5)
	TY	7	0 (0.0)	0 (0.0)
Manila clam	BS	15	0 (0.0)	1 (6.7)
(*Venerupis philippinarum*)	SS	8	1 (12.5)	3 (37.5)
Granular ark (*Tegillarca garnosa*)	BG	14	0 (0.0)	4 (28.6)
Venus clam (*Mercenaria mercenaria*)	BA	12	1 (8.3)	1 (8.3)
Common orient clam (*Meretrix meretrix*)	MS	5	0 (0.0)	1 (20.0)
Scallop (*Patinopecten yessoensis*)	TY	6	0 (0.0)	1 (16.7)
Total		147	5 (3.4)	23 (15.6)

* BA, Buan; BG, Beolgyo; BS, Busan; DC, Daecheon; GJ, Geoje; GS, Gosung; MS, Masan; SS, Seosan; TY, Tongyeong. ** No. of positive groups/No. of total analyzed groups × 100.

**Table 4 animals-13-03348-t004:** Prevalence of white spot syndrome virus (WSSV) entrapped in shellfish imported from China and Japan.

Samples	No. of Groups	No. of PCR Positives (%)	
		First-Step PCR	Second-Step PCR
China			
Manila clam (*Venerupis philippinarum*)	13	0 (0.0) *	0 (0.0)
Venus clam (*Mercenaria mercenaria*)	6	1 (16.7)	1 (16.7)
Common orient clam (*Meretrix meretrix*)	4	0 (0.0)	1 (25.0)
Chinese cyclina (*Cyclina sinensis*)	3	0 (0.0)	1 (33.3)
Scallop (*Pationopecten yessoensis*)	2	0 (0.0)	0 (0.0)
Purple washington clam (*Saxidomus purpurata*)	1	0 (0.0)	0 (0.0)
Bittersweet clam (*Glycymeris vestita*)	1	0 (0.0)	0 (0.0)
Sub total	30	1 (3.3)	3 (10.0)
Japan			
Granular ark (*Tegillarca granosa*)	8	0 (0.0)	1 (12.5)

* No. of positive groups/No. of total analyzed groups × 100.

## Data Availability

The datasets used in this study are available from the corresponding authors upon reasonable request.

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
