# Peer review of "Identification of Various InDel-II Variants of the White Spot Syndrome Virus Isolated from Frozen Shrimp and Bivalves Obtained in the Korean Commercial Market"

_animals, 2023, doi:10.3390/ani13213348_

Round 1
Reviewer 1 Report
The authors have conducted testing of retail collected samples (local and imported) for the detection of WSSV. Positive samples have been further analysed using primers which detect variant in ORF23/24. The study is of interest to the shrimp virology community however the analysis and discussion could be improved. The written grammar and described methodology of the report is of a high standard however the authors could provide a more thorough analysis of the positive samples (Refer Sup Table 1: specifically why are some of the samples positive by gel PCR yet not detected or not tested by qPCR?). Suggest those samples that were not tested by qPCR which were positive by gel PCR should be tested to determine the copy number. Very low copy number may be the reason for non-amplification of ORF23/23 PCRs. Additional analysis aside, the authors could prepare more substantiative figure comparing their analysis with those who have previously analyzed ORF23/24. For example, the inclusion of the ORF23/24 data results of Zwart et al would allow the reader direct comparison of the current study to the previous studies. Such a comparison indicates this study has identified further variation in ORF23/24 to that previously described. Presently the article describes variation in the Korea retail sourced samples but does not present the results of the analysis within a global context of WSSV. Although the article is of high quality, adding a more substantial comparison of the results with those previously conducted would increase the significance of the study and increase the interest to the readers score and overall merit of the article.
Author Response
# Reviewer 1
Q: The authors have conducted testing of retail collected samples (local and imported) for the detection of WSSV. Positive samples have been further analysed using primers which detect variant in ORF23/24. The study is of interest to the shrimp virology community however the analysis and discussion could be improved. The written grammar and described methodology of the report is of a high standard however the authors could provide a more thorough analysis of the positive samples (Refer Sup Table 1: specifically why are some of the samples positive by gel PCR yet not detected or not tested by qPCR?). Suggest those samples that were not tested by qPCR which were positive by gel PCR should be tested to determine the copy number. Very low copy number may be the reason for non-amplification of ORF23/23 PCRs. Additional analysis aside, the authors could prepare more substantiative figure comparing their analysis with those who have previously analyzed ORF23/24. For example, the inclusion of the ORF23/24 data results of Zwart et al would allow the reader direct comparison of the current study to the previous studies. Such a comparison indicates this study has identified further variation in ORF23/24 to that previously described. Presently the article describes variation in the Korea retail sourced samples but does not present the results of the analysis within a global context of WSSV. Although the article is of high quality, adding a more substantial comparison of the results with those previously conducted would increase the significance of the study and increase the interest to the readers score and overall merit of the article.
Response: We would like to thank you for evaluating our manuscript and for the insightful comments.
First, we agree with your thoughts on qPCR, and according to your suggestion, we conducted q-PCR for the non-tested samples (Table S1 and Lines 236–242 in Results), and added a discussion concerning the discrepancies between PCR and qPCR results (Discussion, lines 385–394). In contrast, we conducted qPCR and identified an amplification ranging from 5.92 × 102 to 8.86 × 104 WSSV genome copies/mg (mean, 2.39 × 104 WSSV genome copies/mg) from domestic bivalve samples, and one sample showed amplification with a genome copy value of 4.17 × 104 WSSV genome copies/mg in Chinese Venus clams. Nonetheless, there are instances, in which PCR can be used, but qPCR cannot. We have addressed this matter in the Discussion section (lines 385–394), but in summary, it is attributed to the detection limit difference between the second-step PCR and qPCR (Kim et al., 2023). Additionally, based on your mention of the non-amplification of ORF 23/24 PCRs, we have incorporated this information (Lines 450–453 in the Discussion section) and included a similar reference material to support it (Onihary et al., 2021).
Second, as per your comments, we compared ORF 23/24 variants reported in previous studies. We have revised the analyzed data for ORF 23/24 with the available sequences from GenBank and applied bold formatting to distinguish the variants identified in this study from those identified in a previous research (Figure 2). Additionally, we denoted the sources of the WSSV variants (domestic and imported shrimp and bivalves) by adding footnotes (a–d) to each sample code. Also, we have revised the article to include a more comprehensive global context. The revised manuscript provides a more thorough analysis of WSSV variation, considering both South Korean retail samples and their implications within the broader global context of WSSV (lines 428–433, 436–453 in the Discussion section).
Additionally, we sought the guidance of an expert to make comprehensive improvements to our manuscript, striving to make it more concise and lucid. This collaborative effort aimed to enhance the overall quality of the document and to ensure that our research findings are effectively and clearly communicated.
Thank you for your valuable feedback with your review comments. We believe that the revised manuscript is logically structured and conveys the meaning more simply and clearly. If you have further feedback or suggestions, please feel free to share it. Thank you for your valuable insight.
[Reference]
Kim, M. J., Kim, J. O., Jang, G. I., Kwon, M. G., & Kim, K. I. (2023). Diagnostic validity of molecular diagnostic assays for white spot syndrome virus at different severity grades. Heliyon, 9(9).
Onihary, A. M., Razanajatovo, I. M., Rabetafika, L., Bastaraud, A., Heraud, J. M., & Rasolofo, V. (2021). Genotype Diversity and Spread of White Spot Syndrome Virus (WSSV) in Madagascar (2012–2016). Viruses, 13(9), 1713.

Reviewer 2 Report
Dear authors,
I suggest a more extensive bibliographic review to support the introduction and a brief review of the English wording in the final part of the discussion, observing mainly the assertions of its results, which I consider notable.
Done, above.
Author Response
Q: I suggest a more extensive bibliographic review to support the introduction and a brief review of the English wording in the final part of the discussion, observing mainly the assertions of its results, which I consider notable.
Response: We would like to thank you for evaluating our manuscript and for the insightful comments. In response to your comments, we have conducted an in-depth bibliographic review to enhance the content in the Introduction (Lines 60–67) and carefully revised the English wording in the Discussion to better emphasize the significance of our results (Lines 466–478 in discussion section). In conclusion sections, we also reflected on the implications of our research findings and subsequently provided an in-depth discussion of their significance (Line 488-492 in conclusion section). Additionally, we sought the guidance of an expert to make comprehensive improvements to our manuscript, striving to make it more concise and lucid. This collaborative effort aimed to enhance the overall quality of the document and to ensure that our research findings are effectively and clearly communicated. Thank you for your thoughtful guidance, and we are committed to delivering an improved manuscript.

Reviewer 3 Report
The manuscript titled “Identification of different InDel-II variants of white spot syndrome virus from frozen shrimp and bivalves in the Korean commercial market” presents interesting information on the InDel-II region variability of WSSV from local farms in Korea and other regions. In addition, bivalve species are also evaluated. The manuscript is fairly well written, for the most part, with some paragraphs being problematic (identified below). It is fundamental that the authors change 1st and 2nd PCR for nested-PCR first and second step. An extremely interesting and relevant result is that infected shrimp from other regions that were positive failed to transmit the infection. In order for this manuscript to be published I strongly suggest making all sequencing data available through GenBank or a similar database. Considering these observations and once these minor queries have been address I recommend this manuscript for publication.
Additional comments include:
· I strongly suggest changing the title to: Identification of various InDel-II variants of the white spot syndrome virus isolated from frozen shrimp and bivalves obtained in the Korean commercial market.
· In the simple summary please use the acronym for white spot syndrome virus (WSSV).
· Lines 41-42. Change to “belong to the family Nimaviridae”
· Line 62. Change to “only in wild fish populations”
· Lines 61-64 are hard to follow and this sentence needs to be rewritten.
· Line 86-87. Delete “and its genetic correlation” or at least give more insight of what this means.
· Lines 87-94. Please clarify what groups mean. Are groups a pool of organism? Or are they individual organism? How many shrimp/bivalves are in each group?
· Line 103-104. Change to “ For WSSV detection, a nested-PCR protocol was designed that targets the VP28 gene, for the first step primers VP28F1/R1 were utilized while for the second step primers WSSV VP28-F2-R2 were used”.
· Line 112. Delete “for PCR”.
· Line 183-184. This sentence belongs in the results.
· Line 197. Please correct the font discrepancy.
· Line 210-211. This sentence is confusing and needs to be rewritten.
· Line 212. Change to “While 3.45% of imported samples were PCR-positive”.
· Over all lines 210-218 are not well written and must be rewritten for clarity.
· Line 246. Indicates that a t-test was used for copy number detection. Please describe the t-test and data normalization in the material and methods.
· Line 247-262. Since the sequence of each of these variants is available. It should be uploaded to GenBank and the accession numbers must be provided. In addition, please indicate if any sequence variation was detected in the form of SNPs.
· Line 293-295. This sentences is not clear please clarify. Do you mean this study helps assess the potential transmission from frozen shrimp? How is that related to molecular epidemiology?
·
Author Response
Q: The manuscript titled “Identification of different InDel-II variants of white spot syndrome virus from frozen shrimp and bivalves in the Korean commercial market” presents interesting information on the InDel-II region variability of WSSV from local farms in Korea and other regions. In addition, bivalve species are also evaluated. The manuscript is fairly well written, for the most part, with some paragraphs being problematic (identified below). It is fundamental that the authors change 1st and 2nd PCR for nested-PCR first and second step. An extremely interesting and relevant result is that infected shrimp from other regions that were positive failed to transmit the infection. In order for this manuscript to be published I strongly suggest making all sequencing data available through GenBank or a similar database. Considering these observations and once these minor queries have been address I recommend this manuscript for publication.
Response: We would like to thank you for evaluating our manuscript and for the insightful comments.
First, as per your comments, we have revised the terms “1st and 2nd PCR” to “first- and second- PCR” throughout the entire manuscript.
Second, the failure of WSSV to transmit infection using imported (or from other regions) shrimp can be attributed to several factors. Although WSSV has been shown to maintain its pathogenicity when stored in a frozen state for 2 years (Aranguren et al., 2020; Lightner et al., 1997), there is a possibility of a relatively long storage duration during which virulence may decrease over time. Furthermore, lower viral copy numbers in imported shrimp may play a role in the failure of WSSV transmission. This observation corresponds with the study by Kim et al. (2023), suggesting that WSSV copy numbers below 104.93 genome copies/mg are insufficient to induce mortality, indicating that the infection levels in imported shrimp may not have been high enough to cause mortality. Based on your comments, we have incorporated this information (lines 371–381). In light of your feedback and consideration, we have revised our manuscript to ensure the effective communication of our findings.
Third, we agree with your comments and have submitted all sequencing data to GenBank. We have submitted our sequence to GenBank (Banklt 2752446). However, there has been a delay in receiving the accession number due to minor issues. When we receive accession no. from GenBank, we will incorporate this information in the research article.
Additionally, we sought the guidance of an expert to make comprehensive improvements to our manuscript, striving to make it more concise and lucid. This collaborative effort aimed to enhance the overall quality of the document and to ensure that our research findings are effectively and clearly communicated.
Thank you for your valuable feedback with your review comments. We believe that the revised manuscript is logically structured and conveys the meaning more simply and clearly. If you have further feedback or suggestions, please feel free to share it. Thank you for your valuable insight.
Q2. I strongly suggest changing the title to Identification of various InDel-II variants of the white spot syndrome virus isolated from frozen shrimp and bivalves obtained in the Korean commercial market.
Response: Thank you for your comments. Accordingly, we have revised our manuscript title from “Identification of different InDel-II variants of white spot syndrome virus from frozen shrimp and bivalves in the Korean commercial market” to “Identification of various InDel-II variants of the white spot syndrome virus isolated from frozen shrimp and bivalves obtained from the Korean commercial market” to clarify the meaning of the title.
Q3. In the simple summary please use the acronym for white spot syndrome virus (WSSV).
Response: Thank you for your comments. We have used the acronym “WSSV” in the revised manuscript.
Q4. Lines 41-42. Change to “belong to the family Nimaviridae”
Response: Thank you for your comments. We have changed “belong to Nimaviridae” to “belong to the family Nimaviridae” according to your comment (Line 40).
Q5. Line 62. Change to “only in wild fish populations”, Lines 61-64 are hard to follow and this sentence needs to be rewritten.
Response: Thank you for your comments. Based on your comments, we have revised this part (Lines 60–67).
Q6. Line 86-87. Delete “and its genetic correlation” or at least give more insight of what this means.
Response: Thank you for your comments. Based on your feedback, we have revised the description to clarify its meaning (Lines 91–96).
Q7. Lines 87-94. Please clarify what groups mean. Are groups a pool of organism? Or are they individual organism? How many shrimp/bivalves are in each group?
Response: Thank you for your comments. According to your advice, we have clarified the meaning of “groups” as a pooling of individual animals in the text. Additionally, we have included the number of shrimp or bivalves pooled in each group, as per your suggestion (Lines 91–96).
Q8. Line 103-104. Change to “ For WSSV detection, a nested-PCR protocol was designed that targets the VP28 gene, for the first step primers VP28 F1/R1 were utilized while for the second step primers WSSV VP28-F2/R2 were used”
Response: Thank you for your comments. Based on your comment, we have revised the sentence to ensure clarity and accuracy (Lines 113–115)
Q9. Line 112. Delete “for PCR”.
Response: Thank you for your comment. We have removed the redundant information
Q10. Line 183-184. This sentence belongs in the results.
Response: Thanks for your advice. We have relocated the sentence to an appropriate section in the Results (Lines 225–228).
Q11. Line 197. Please correct the font discrepancy.
Response: Thank you for your comments. We have corrected this error accordingly.
Q12. Line 210-211. This sentence is confusing and needs to be rewritten., Over all lines 210-218 are not well written and must be rewritten for clarity. Line 212. Change to “While 3.45% of imported samples were PCR-positive”
Response: Thank you for your valuable comments. Based on your feedback, we have enhanced the description of the WSSV detection results, aiming for greater clarity (Lines 214–223)
Q13. Line 246. Indicates that a t-test was used for copy number detection. Please describe the t-test and data normalization in the material and methods.
Response: Thank you for your advice. Following your suggestion, we have added more information (Lines 168–171).
Q14. Line 247-262. Since the sequence of each of these variants is available. It should be uploaded to GenBank and the accession numbers must be provided. In addition, please indicate if any sequence variation was detected in the form of SNPs.
Response: Thank you for your valuable comments. As described previously, we submitted the sequencing data to GenBank. These results will be incorporated as soon as they become available. Among the WSSV variants found in bivalves, previously unreported SNPs were observed in the two WSSV variants (12-BS-Mu-9 and 14-GJ-Mu-2). Therefore, we have incorporated the findings related to SNPs into our manuscript and also discussed the implications of these discoveries (Line 441-450 in Discussion section).
Q 15. Line 293-295. This sentences is not clear please clarify. Do you mean this study helps assess the potential transmission from frozen shrimp? How is that related to molecular epidemiology?
Response: Thank you for your comments. According to your question, we have improved the explanation to convey the significance of the molecular epidemiological analysis more clearly (Lines 345–362).
[Reference]
Aranguren Caro, L. F., Mai, H. N., Nunan, L., Lin, J., Noble, B., & Dhar, A. K. (2020). Assessment of transmission risk in WSSV‐infected shrimp Litopenaeus vannamei upon cooking. Journal of fish diseases, 43(4), 403-411.
Lightner, D. V., Redman, R. M., Poulos, B. T., Nunan, L. M., Mari, J. L., & Hasson, K. W. (1997). Risk of spread of penaeid shrimp viruses in the Americas by the international movement of live and frozen shrimp. Revue scientifique et technique (International Office of Epizootics), 16(1), 146-160.
Kim, M. J., Kim, J. O., Jang, G. I., Kwon, M. G., & Kim, K. I. (2023). Evaluation of the Horizontal Transmission of White Spot Syndrome Virus for Whiteleg Shrimp (Litopenaeus vannamei) Based on the Disease Severity Grade and Viral Shedding Rate. Animals, 13(10)

Reviewer 4 Report
The manuscript investigated the genetic relatedness and pathogenicity of four different WSSV variants from frozen shrimp and shellfish. This study can be helpful to reveal a link between InDel-II deletions and viral replication of WSSV variants. But there are several serious problems in the manuscript. In this manuscript, there are few data on the genetic relatedness and pathogenicity of the four variants, only the results of challenge tests are not sufficient to support the conclusions of the entire article. It is recommended that add more comprehensive validation experiments. The Figure 3 as an important result of the manuscript, the figure and caption are not clear to show the results clearly.
Author Response
Q: The manuscript investigated the genetic relatedness and pathogenicity of four different WSSV variants from frozen shrimp and shellfish. This study can be helpful to reveal a link between InDel-II deletions and viral replication of WSSV variants. But there are several serious problems in the manuscript. In this manuscript, there are few data on the genetic relatedness and pathogenicity of the four variants, only the results of challenge tests are not sufficient to support the conclusions of the entire article. It is recommended that add more comprehensive validation experiments. The Figure 3 as an important result of the manuscript, the figure and caption are not clear to show the results clearly.
Response: We would like to thank you for evaluating our manuscript and for the insightful comments.
In accordance with your comments, we have conducted additional experiments to investigate the differences in viral replication during the early stages of WSSV infection. To address this, we injected the Kr-1 and Kr-4 isolates, which showed significant differences in pathogenicity in previous experiments, at the same concentrations (106 genome copies/shrimp). Additionally, we compared viral copy values in shrimp tissues at hours post injection (hpi) +12, 24, 48, and 72 h (Lines 199-206 in Materials and methods). This experiment was conducted to assess the dynamics of the infection during the early infection stages. In the results, despite the same injection concentration (106 genome copies/shrimp), we observed differences in WSSV viral copies values within shrimp tissues at hpi +12 and +24. Importantly, these differences were found to be statistically significant with p-value of 0.0099 and 0.0092, respectively (Figure 4). This suggests that variations can lead to significant differences in virus replication during the early stages of infection, even the initial inoculum copies values are the same. Furthermore, in response to your feedback, we have revised the caption of Figure 3 to clearly show the results.
Additionally, we sought the guidance of an expert to make comprehensive improvements to our manuscript, striving to make it more concise and lucid. This collaborative effort aimed to enhance the overall quality of the document and to ensure that our research findings are effectively and clearly communicated.
Thank you for your valuable feedback with your review comments. We believe that the revised manuscript is logically structured and conveys the meaning more simply and clearly. If you have further feedback or suggestions, please feel free to share it. Thank you for your valuable insight.

Reviewer 5 Report
In the manuscript “Identification of different InDel-II variants of white spot syndrome virus from frozen shrimp and bivalves in the Korean 3 commercial market”, the authors demonstrate the presence of white spot syndrome virus from shrimp and bivalves in Korean markets. They authors go on to show a variety of deletion variants in the 23/24 genomic region. The authors also demonstrate that larger deletions accelerate mortality.
Specific comments about the manuscript include:
1. Line 172, should use ng DNA vs μL of DNA
2. In the discussion, there should be some references about 1st vs 2nd round PCR. Is this standard in terms of analysis and when looking at the 2nd round, what are the chances for false positives, due to things like small amounts of contaminating DNA.
3. In addition, to a discussion of two rounds of PCR, what controls were used to prevent cross contamination – should be included in M&M. Particularly around 2nd PCR run.
4. Figure 3 – a different symbol or dashed line should be used for KR-3. Because of the shape, KR-3 doesn’t appear on the graph as its symbol is covered by symbols used for other samples.
5. Does the deletion within 23/24 create any new ORFs or alter existing ORFs? This should be discussed.
6. What does 23/24 ORFs do or proposed to do?
Author Response
Q1. Line 172, should use ng DNA vs μL of DNA
Response: We would like to thank you for evaluating our manuscript and for the insightful comments.
We have revised the manuscript to include microliters of DNA. (Line 160 in Materials and Methods section)
Q2. In the discussion, there should be some references about 1st vs 2nd round PCR. Is this standard in terms of analysis and when looking at the 2nd round, what are the chances for false positives, due to things like small amounts of contaminating DNA.
Response: Thank you for your valuable comments. We have incorporated references and discussions regarding the first and second round of PCR. Briefly, a positive result in the first PCR typically indicates a serious infection stage, whereas a positive result in the second PCR examination indicates a latent or carrier-state infection (Lines 353–356). Regarding false positives, as highlighted by Claydon et al. (2004), to avoid cross-contamination in the second round of PCR, WSSV-free shrimp tissues were utilized as a negative control during the sample analysis process (Lines 140–142 in the Materials and Methods section). In addition, we have improved the discussion concerning the false positives to make it clearer. These additions have enhanced the overall quality of the discussion and provided a more comprehensive analysis of the methodology. Thank you for this valuable suggestion.
Q3. In addition, to a discussion of two rounds of PCR, what controls were used to prevent cross contamination – should be included in M&M. Particularly around 2nd PCR run.
Response: Thank you for your valuable comments. We have incorporated a section in the Materials and Methods section to present the controls used to prevent cross-contamination, especially during the second-step PCR run (Lines 140-142 in the Materials and Methods section).
Q4. Figure 3 – a different symbol or dashed line should be used for KR-3. Because of the shape, KR-3 doesn’t appear on the graph as its symbol is covered by symbols used for other samples.
Response: Thank you for your comment. We have revised the figure to ensure that all the variants are visually distinguishable from the other samples. This adjustment should resolve the issues you raised. Thank you for your detailed consideration.
Q5. Does the deletion within 23/24 create any new ORFs or alter existing ORFs? This should be discussed.
Response: We appreciate your inquiry into the potential effects of deletion in the ORF 23/24 region. Owing to the presence of the InDel-II region within the ORF 23/24 region of the ancestral “WSSV-TH isolate,” it was initially referred to as the ORF 23/24 region. However, more recent studies have proposed multiple ORFs within this region (Zwart et al., 2010, and Hoa Hoa et al. (2012) suggested the existence of a greater variety of ORFs in the corresponding regions of Taiwanese or Chinese isolates. Considering these aspects, insertions or deletions in the InDel region may result in variations in the ORFs present in WSSV, potentially resulting in differences in pathogenicity. We have added this information to the Discussion believing that it will be helpful in providing meaningful insights to the readers. Thank you for your feedback.
Q6. What does 23/24 ORFs do or proposed to do?
Response: Thank you for your question. In a previous study, ORF23/24, which represented an InDel-II region with an identified deletion length of approximately 13 kb, served as a significant genetic marker for epidemiological tracking (Zwart et al., 2010). In particular, the InDel-II region has been utilized to investigate the transmission pathways of WSSV between regions and on a global scale (Dieu et al., 2010). Based on this research, we considered “the ORF 23/24” as a critical tool for our investigation of the WSSV transmission pathway and conducted our analysis using shrimp and bivalves. Therefore, this region can be used for the epidemiological tracking and identification of WSSV types distributed in endemic regions. Furthermore, it enables comparative studies of the pathogenicity in these areas. Based on your question, we have revised the corresponding part in the manuscript (Lines 488–492 in Conclusions section) and improved it to provide clarity on this aspect of the experiment. Thank you for your valuable comments.
Overall, we sought the guidance of an expert to make comprehensive improvements to our manuscript, striving to make it more concise and lucid. This collaborative effort aimed to enhance the overall quality of the document and to ensure that our research findings are effectively and clearly communicated.
We appreciate your kind words regarding our manuscript. If you need assistance with any questions or help on topics such as paper writing or any other matter, please don’t hesitate to reach out.
[Reference]
Claydon, K., Cullen, B., & Owens, L. (2004). OIE white spot syndrome virus PCR gives false-positive results in Cherax quadricarinatus. Diseases of aquatic organisms, 62(3), 265-268
Dieu, B. T. M., Marks, H., Zwart, M. P., & Vlak, J. M. (2010). Evaluation of white spot syndrome virus variable DNA loci as molecular markers of virus spread at intermediate spatiotemporal scales. Journal of General Virology, 91(5), 1164-1172.
Hoa, T. T. T., Zwart, M. P., Phuong, N. T., Oanh, D. T., de Jong, M. C., & Vlak, J. M. (2012). Indel-II region deletion sizes in the white spot syndrome virus genome correlate with shrimp disease outbreaks in southern Vietnam. Diseases of Aquatic Or-ganisms, 99(2), 153-162.
Zwart, M. P., Dieu, B. T. M., Hemerik, L., & Vlak, J. M. (2010). Evolutionary trajectory of white spot syndrome virus (WSSV) genome shrinkage during spread in Asia. PLoS One, 5(10), e13400.

Round 2
Reviewer 4 Report
The authors carefully revised the manuscript. In my personal opinion, the revised manuscript is acceptable. Only in line 205-207, Two-way ANONA, or One-way ANONA?
Author Response
Q: The authors carefully revised the manuscript. In my personal opinion, the revised manuscript is acceptable. Only in line 205-207, two-way ANOVA, or One-way ANOVA?
Response: We appreciate for your valuable feedback.
In our previous additional experiments aimed at comparing the differences in pathogenicity among variants, we initially conducted a Two-way ANOVA analysis to assess the difference in WSSV genome copy values using two variants within the ‘same time frame’. However, as you rightly pointed out, it seems like One-way ANOVA test lead to confusion. To mitigate this, we carried out an analysis involving comparisons among all groups, thus opting for Two-way ANOVA. The outcomes of this analysis have been incorporated into Figure 4, allowing for a more comprehensive examination of the results. And we also updated the values in the manuscript to reflect the changes in the outcomes (Line 308-309, 311-312, 326-328 in results section, Line 465 in discussion sections)
Once again, I am sincerely thankful for your constructive feedback.
